# Methylphenidate Use for Emotional Dysregulation in Children and Adolescents with ADHD and ADHD and ASD: A Naturalistic Study

**DOI:** 10.3390/jcm11102922

**Published:** 2022-05-22

**Authors:** Patrizia Ventura, Concetta de Giambattista, Paolo Trerotoli, Maddalena Cavone, Alessandra Di Gioia, Lucia Margari

**Affiliations:** 1Child Neuropsychiatric Unit, University of Bari Aldo Moro, Piazza Giulio Cesare, 70121 Bari, Italy; patriziaventuranpi@gmail.com (P.V.); concettadegiambattista@gmail.com (C.d.G.); magda.cavone@gmail.com (M.C.); aledigioia1993@gmail.com (A.D.G.); 2Department of Biomedical Sciences and Human Oncology, University of Bari, 70010 Bari, Italy; paolo.trerotoli@uniba.it

**Keywords:** methylphenidate, MPH, emotional dysregulation, ED, autism spectrum disorder, ASD, attention-deficit/hyperactivity disorder, ADHD

## Abstract

Emotional dysregulation (ED) is common in attention-deficit/hyperactivity disorder (ADHD). Nonetheless, research on ADHD in children with autism spectrum disorder (ASD) and ADHD is still ongoing. Several studies suggest that methylphenidate (MPH) may be effective for ED in ADHD, while there is not enough evidence about its use in ASD with comorbid ADHD. This naturalistic study aims to investigate the effectiveness of immediate- and extended-release MPH in the treatment of ED in 70 children and adolescents (6–18 years), with a diagnosis of ADHD (*n* = 41) and of ASD with comorbid ADHD (*n* = 29), using the Child Behavior Checklist—Attention/Aggressive/Anxious (CBCL-AAA). Their parents completed the CBCL twice—first during the summer medication-free period, that is, at least one month after drug interruption; and again after three months of treatment restart. Results demonstrate that MPH is associated with a statistically significant reduction in ED in ADHD and ASD, without substantial adverse events, supporting the use of psychostimulants for the treatment of ED in these neurodevelopmental disorders.

## 1. Introduction

Emotional dysregulation (ED) is characterized by an inability to modulate emotional responses, resulting in extreme reactions of an internalizing or externalizing nature, inappropriate for developmental age [1]. It has long been recognized that ED is common in individuals with neurodevelopmental disorders, including attention-deficit/hyperactivity disorder (ADHD) and autism spectrum disorder (ASD) [2,3,4,5]. ADHD is defined by age-inappropriate inattention, hyperactivity, and impulsivity and has a prevalence of 5% (2–9.4% [6,7,8]; 12.9% in boys, 5.6% in girls [8]). Individuals with ADHD often present excessive and rapidly shifting emotions, associated with irritable and aggressive behavior [9], susceptibility to anger and low tolerance for distress [10,11,12,13,14,15,16,17,18]. These symptoms are more prevalent in the ADHD combined subtype, especially when comorbid with oppositional defiant disorder [19], and their severity increases with the severity of other ADHD symptoms [4]. Individuals with ADHD combined with ED were more impaired in global functioning [20]. ASD is defined by persistent deficits in social interaction and communication, as well as restricted, stereotyped and repetitive behaviors, and has an overall prevalence between 1 and 2.93% [21,22,23]. Even if ED is not a formal criterion for the diagnosis of ASD, emotional and behavioral problems, including irritability, temper outbursts, anxiety, aggression, and self-injury are frequently observed in ASD [5,24]. Research shows that, when compared to typically developing individuals, people with ASD are generally less effective or maladaptive at using emotional regulation strategies [2,25,26,27].

As for therapy, to date, no data are available regarding a specific pharmacological treatment for ED. To date, in most cases, treatment choice was guided by the main psychiatric condition behind ED. In ADHD individuals, preliminary evidence suggests that psychostimulants improve emotional recognition and reduce emotional lability and irritability [28]. Nevertheless, most studies demonstrated the effect of MPH treatment on ED in patients with uncomplicated ADHD [28,29,30,31,32,33,34,35]. There have been far few studies investigating the efficacy and safety of MPH in those with ADHD and co-existing disorders: some studies concluded that MPH remains effective at reducing ADHD symptoms in the presence of affective comorbidities, without an effect on comorbid conditions (anxiety, depression [36]); other studies suggested that MPH treatment can also reduce other emotional and behavioral problems (such as aggression and antisocial behavior [37]), mood lability [30] and obsessive and compulsive symptoms [32]. Indeed, there is a lack of studies about the effect of psychostimulants on ED in ADHD individuals with coexisting ASD.

The aim of this study was to evaluate the effectiveness and tolerability of immediate- and extended-release MPH treatment on ED in children and adolescents with ADHD, with or without ASD.

## 2. Materials and Methods

### 2.1. Participants and Procedures

This was a naturalistic study based on a clinical database of Caucasian young people, aged between 6 and 18 years, consecutively referred to the Child Neuropsychiatric Unit of the University of Bari during a two-year period (September 2019–September 2021). The inclusion criteria were a diagnosis of moderate to severe ADHD, according to DSM-5 [38], pharmacological treatment with immediate-release MPH (IR-MPH) or extended-release MPH (ER-MPH) and therapy lasting at least three months. Given the naturalistic design of this study, patients who already took other pharmacological treatments were not excluded. All the subjects were diagnosed according to clinical judgement from the expert team, composed of child and adolescent neuropsychiatrists and psychologists, specialists in neurodevelopmental disorders. Diagnosis was also supported by standardized diagnostic tools, specific for ADHD and ASD: the Revised Conners’ Parent Rating Scale (CPRS-R) [39], the Autism Diagnostic Observational Schedule, Second Edition (ADOS-2) [40,41], the Autism Diagnostic Interview-Revised (ADI-R) [42] and the Autism Spectrum Diagnostic Interview (ASDI) [43]. Moreover, the intellectual quotient of all subjects was assessed with Wechsler scales: the Wechsler Preschool and Primary Scale of Intelligence-Third Edition (WPPSI-III) [44] and the Wechsler Intelligence Scales for Children-Fourth Edition (WISC-IV) [45], or the Leiter International Performance Scale-Revised (Leiter-R) [46] in patients with communication impairment. Subjects were also screened for other psychiatric disorders using medical history, clinical observation and the Child Behavior Checklist (CBCL) [47]. All the therapeutic procedures, as well as the follow-up and data collection, were part of our standard routine. All subjects and parents received detailed information on different treatment options and gave their written informed consent to the treatment with MPH. Medication was prescribed as clinically indicated, in addition to routine community care (including psychoeducational interventions), as part of a management protocol.

The starting dose of MPH was 0.3–0.5 mg/kg/day. The dosage could be increased up to 1 mg/kg/day depending upon the subject’s clinical response and tolerability, up to a maximum of 60 mg/day. The total dose could be administered in two or three doses/day. After one month of titration, the IR-MPH was generally replaced with ER-MPH. The therapeutic management of ADHD children in our unit comprises short breaks in summer, in agreement with parents and patients about the best pattern of use, with the aim of investigating the ongoing benefit of medication, helping to verify need and the possibility of a reduction in side effects relating to sleep and appetite, albeit considering its reintroduction whenever needed. The median MPH dose and therapy duration are described in Table 1.

For the specific purpose of this study, ED was studied through the evaluation of CBCL scores.

The CBCL is a self-reported scale developed by Achenbach and Edelbrock to evaluate existing competencies and behavioral, emotional, and social problems in children and adolescents aged between 6 and 18 years, through information derived from parents. The behavioral problems scales include 118 items, divided in internalizing and externalizing subscales. The internalizing subscale contains items concerning withdrawn behavior, somatic complaints, and anxious/depressed behavior. The externalizing subscale includes items concerning delinquent and aggressive behavior. Furthermore, there are subscales that contain social, thought, and attention problems. The higher score represents a higher level of problems. A computer program calculates the t scores (mean 50, standard deviation 10) for each scale. Raw scores are converted to gender- and age-standardized scores.

According to Biederman [13,48,49,50,51], ED was defined as positive if the sum of the CBCL “attention problems”, “aggressive behavior”, and “anxious-depressed” (CBCL-AAA) t scores was equal or higher than 180. Moreover, it is possible to distinguish two different profiles: (1) CBCL-DESR (Deficient Emotional Self-Regulation) when t scores are between 180 and 210; (2) CBCL-SED (Severe Emotional Dysregulation) when t scores are higher than 210. When scores are lower than 180, no profile is developed (no-ED). In order to evaluate the effects of MPH therapy on ED, the research team analyzed the CBCL scores obtained during the summer medication-free period, at least one month after drug interruption (T0), and after three months of treatment restart (T1).

Data obtained from the ADHD group and the ADHD and ASD group were compared to each other, henceforth analyzed.

### 2.2. Ethic Considerations and Sampling

The Ethic Committee of the Azienda Ospedaliera-Universitaria Consorziale Policlinico di Bari approved the study. Informed consent from parents and assent from children and adolescents were obtained prior to enrollment. Among 75 patients who were prescribed MPH, 5 were excluded because their caregiver did not sign the informed consent. The enrolled sample consisted of 70 patients of which 41 had a diagnosis of ADHD (henceforth named the ADHD group) and 29 had a diagnosis of ASD with comorbid ADHD (henceforth named the ADHD and ASD group).

### 2.3. Statistical Analysis

Comparisons between the two groups were analyzed by t-test or Wilcoxon test for independent samples, appropriately selected according to the evaluation of normality of quantitative variables. Comparisons of proportions between independent groups have been performed by the chi-square test or the Fisher exact test, according to appropriateness. Comparisons between proportions in paired samples were compared by the McNemar test. A general linear model was also applied to evaluate the difference between the two groups at the two timepoints. The main aim was to evaluate whether there was a difference between the two groups (ADHD vs. ADHD and ASD) in terms of effectiveness and tolerability of MPH on ED; therefore, a test of equivalence was performed: the Two One Side Test (TOST) was applied comparing the limits of the 90% confidence interval of the difference between the two means with the of equivalence (D). They were determined as the product between the effect size (ES) (fixed at 0.5 according Cohen’s medium *es*) and the pooled standard deviation: D = *es* × S_p_.

All the analyses were performed using SAS 9.4 for personal computer. The significance level was set at *p* < 0.05. The test of equivalence was performed by the *t*-Test Procedure applying the TOST option with the option Alpha 0.05 (i.e., the TOST will be performed comparing limits of 90% CI of the difference between two means).

## 3. Results

### 3.1. Sociodemographic Characteristics

Demographic and clinical characteristics of the two groups, comorbidities and pharmacological treatments in addition to MPH are described in Table 2.

### 3.2. Effectiveness

#### 3.2.1. CBCL-AAA Profiles

Comparing the ADHD group with the ADHD and ASD group, the difference between CBCL-AAA profiles (DESR, SED and no-ED) at T0 and at T1 was not statistically significant (*p* = 0.08 and *p* = 0.18, respectively). Conversely, comparing CBCL-AAA profiles in T0 and T1 in each group and in the total sample, the difference was statistically significant (*p* < 0.01) (Table 3).

#### 3.2.2. CBCL-AAA Scores

Comparing the ADHD group with the ADHD and ASD group, the difference between each subscale of CBCL-AAA both at T0 and at T1 was not statistically significant (T0 anxious: *p* = 0.32, attention: *p* = 0.42, aggressive: *p* = 0.27; T1 anxious: *p* = 0.31, attention: *p* = 0.04, aggressive: *p* = 0.29). Conversely, comparing the scores of CBCL-AAA subscales in T0 and T1 in each group and in the total sample, the difference was statistically significant (*p* < 0.01).

Comparing the two groups, the difference in CBCL-AAA total scores at T0 and T1 was not statistically significant (*p* = 0.24 and *p* = 0.12, respectively). Conversely, comparing the CBCL-AAA total score in T0 and T1 in each group and in the total sample, the difference was statistically significant (*p* = 0.01) (Table 4).

Evaluating the equivalence between the difference T0–T1 of the CBCL-AAA total scores, the two groups were equivalent to each other.

Conversely, evaluating the equivalence between the CBCL-AAA scores in the two group at T0 and at T1, the two groups were not equivalent to each other.

### 3.3. Tolerability

No severe adverse events were reported; indeed, among these, no cardiovascular events neither suicidal ideation or behaviors were seen (Table 5).

The most frequent side effects in the ADHD group were loss of appetite, abdominal discomfort, and headache (46.34%, 24.39% and 17.07%, respectively), each only temporary, in the first days or weeks of treatment; the same side effects were seen in the ADHD and ASD group (44.83%, 34.48% and 17.24%, respectively). The difference between each side effect in the two groups was not statistically significant. Two ASD patients with comorbid ADHD, both with intellectual disability, of which one was affected by level 3 ASD and the other by level 2 ASD, presented worsening of behavior, with restlessness and increased stereotypes that caused the interruption of treatment and completely resolved after treatment discontinuation.

## 4. Discussion

This study analyzes MPH effectiveness and tolerability on emotional dysregulation (ED) in a naturalistic setting of 70 ADHD patients, of which 29 also had ASD. To the best of our knowledge, this is the first study focusing on the effect of MPH on ED in ASD children and adolescents.

Several studies analyzed the prevalence of ED both in ADHD, finding percentages between 24 and 55% in children and adolescents and 34 to 70% in adults [3,13], and in ASD, finding a percentage between 50% and 82% [5,24,25]. In this study, the prevalence of ED within ADHD subjects is 90.25% (DESR in 58.52%, SED in 31.7%), higher with respect to previous findings, probably because our patients are affected by moderate to severe ADHD; the prevalence of ED within ASD with comorbid ADHD is 68.95% (DESR in 44.82%, SED in 24.13%), lower with respect to previous findings, probably because of a low rate (3.45%) of intellectual disability.

ED is a challenging pharmacological target, and the scientific literature to date does not provide tailored guidelines for its management, probably because most of the available pharmacological trials did not focus on ED per se. Given the transnosographic quality of ED, pharmacological approaches aimed at it are largely guided by the presence of comorbid illnesses: risperidone and aripiprazole in the context of ASD and/or intellectual disability, serotonin reuptake inhibitors in the context of depression or generalized anxiety disorders, and mood stabilizers (lithium and divalproex) in the context of bipolar spectrum disorders [3,52,53,54,55].

Previous research and clinical experience clearly demonstrated that MPH has a beneficial effect on ADHD core symptoms [56,57,58]. Recent studies extend these results also on ED in young patients with ADHD, although most of the evidence concerns uncomplicated ADHD [3,28,29,30,31,33,34,37]. To date, not many studies have investigated the efficacy and safety of MPH in ADHD patients with comorbid disorders, albeit preliminary data demonstrated MPH efficacy on oppositional, aggressive and antisocial behaviors, on mood lability, and on obsessive and compulsive symptoms [28,53,54].

Concerning the use of MPH in ASD patients, efficacy and safety data are controversial: some early studies reported that stimulant medication resulted in increased stereotypical movements in children with ASD or in additional adverse events that may impact negatively on social interaction. However, these results are based on small samples of patients (frequently case report), often comorbid with intellectual disability [59,60,61]. Thus, most recent literature on MPH treatment in children with ASD with comorbid ADHD suggests more positive outcomes [12,62,63,64,65,66], with an ES between 0.54 [67] and 0.67 [68] particularly in the case of average intellectual functioning [69]. Indeed, in our previous study, involving 80 ADHD children and adolescents of whom 40 had high-functioning ASD, long-term MPH treatment resulted in amelioration of global functioning [69]. Although promising results on the efficacy of MPH in ASD individuals have been described, no studies have been conducted on its effect on ED to date.

The present study suggests that MPH is associated with a significant improvement in ED in ADHD children and adolescents with or without ASD. Actually, after MPH use, a statistically significant (*p*< 0.0001) reduction in CBCL-AAA score was found both in the ADHD group and in the ADHD and ASD group. Within these findings, it is remarkable to note that the median of intellectual quotient was 97.2 in the ADHD group and 96.3 in the ADHD and ASD group. As for tolerability, treatment with MPH was well tolerated; there were no serious adverse events, with the main adverse events being temporary loss of appetite, abdominal discomfort, and headache, without significant differences in the two groups.

Only two ASD patients with comorbid ADHD, both with intellectual disability, presented worsening of behavior, with restlessness and increased stereotypes that caused the interruption of treatment and completely resolved after treatment discontinuation.

Previous review studies comparing MPH to placebo in children with ASD [62,68] globally show that it was mildly superior for inattention, irritability and stereotypes, and did not worsen the core symptoms of ASD.

Even though the low number of individuals with intellectual disabilities in our sample did not allow us to draw conclusions about the role of intellectual disability on the effects of MPH, our findings are in accordance with previous studies [64] in which high-functioning ASD was more likely to have a favorable response to MPH treatment in terms of efficacy and tolerability than low-functioning ASD and suggest greater caution in monitoring the effects of MPH in ASD with intellectual disability.

The present study has different strengths: it describes the effectiveness and tolerability of MPH on moderate to severe ADHD and ADHD and ASD patients from a well phenotypically characterized sample (intellectual profile, severity specifiers, diagnostic subtypes, and comorbidities), with a specific focus on ED, assessed through a standardized measure. Being a naturalistic study, medication titration and dose adjustment were possible as necessary, almost like in a real-life setting.

Study limitations include the lack of randomization or blinding and control of other pharmacological/behavioral treatments and the small sample size. The suitability of our study design used to measure ED administering the CBCL during the summer medication-free period and after three months of treatment restart (employing each patient as his/her own control) could be discussed. Future research in this field should involve proper administration of the CBCL before the beginning of treatment and the extension of ED assessment through other measures, as objective as possible and preferably not only parent mediated.

In conclusion, our study shows that MPH treatment is well tolerated and effective in reducing ED in children and adolescents with ADHD who may also have high-functioning ASD. Thus, the routine use of MPH in this group of children and adolescents is recommended.

## Figures and Tables

**Table 1 jcm-11-02922-t001:** MPH treatment.

	ADHD		ADHD and ASD		*p* Value
	Median	IQR	Median	IQR	
**MPH dose (mg/kg/day)**	0.78	0.64–0.93	0.8	0.57–1	0.79
**Therapy duration (months)**	24	3–48	36	3–48	0.32

ADHD—attention-deficit/hyperactivity disorder; ASD—autism spectrum disorder; MPH—methylphenidate; IQR—interquartile Range.

**Table 2 jcm-11-02922-t002:** Demographic and clinical characteristics of the sample, comorbidities, and other psychotropic medications in addition to MPH.

	ADHD Group (*N* = 41)		ADHD and ASD Group (*N*= 29)		*p* Value
	** *N* **	**%**	** *N* **	**%**	
**Sex**					
M	29	70.7	23	79.3	
F	12	29.3	6	20.7	0.42
	**Median**	**IQR**	**Median**	**IQR**	
**Age (years)**	13.4	10–17	13.3	11–16	0.95
**ADHD specifiers**					
Inattentive	4	9.8	3	10.3	
Combined	37	90.2	26	89.7	0.48
	**Median**	**IQR**	**Median**	**IQR**	
**Intelligence quotient (IQ)**	97.2	86–112	96.3	82–112	0.85
**Comorbidities**	** *N* **	**%**	** *N* **	**%**	
ID and limited intellectual functioning	3		3		
LD	31	7.3	19	10.3	0.69
MCD	2	75.6	1	65.5	0.36
Tic disorder	1	4.9	0	3.5	0.77
Mood/Anxiety disorder	3	2.4	1	0	0.64
DMDD	2	7.3	0	3.5	0.51
ODD	8	4.9	1	0	0.04
		18.5		3.5	
**Other psychotropic medications**					
Antipsychotics (FGA and SGA)	4	9.8	2	6.9	1.00
Mood stabilizer	1	2.4	0	0	1.00

ADHD—attention-deficit/hyperactivity disorder; ASD—autism spectrum disorder; ID—intellectual disability; LD—learning disorder; MCD—motor coordination disorder; DMDD—disruptive mood dysregulation disorder; ODD—oppositional defiant disorder; FGA—first-generation antipsychotics; SGA—second-generation antipsychotics.

**Table 3 jcm-11-02922-t003:** Changes in CBCL-AAA profiles over 3 months of MPH treatment in the two groups.

CBCL-AAA Profile	ADHD Group *N* (%)	ADHD and ASD Group *N* (%)	*p* Value	Total Sample
**T0**				
SED	13 (31.7)	7 (24.13)	0.08	20 (28.57)
DESR	24 (58.52)	13 (44.82)		37 (52.86)
Non-ED	4 (9.75)	9 (31.03)		13 (18.57)
**T1**				
SED	4 (9.76)	3 (10.34)	0.18	7 (10)
DESR	17 (41.46)	6 (20.69)		23 (32.86)
Non-ED	20 (48.78)	20 (68.97)		40 (57.14)
***p*-value**	<0.01	<0.01		<0.01

ADHD—attention-deficit/hyperactivity disorder; ASD—autism spectrum disorder; CBCL-AAA—Childhood Behavior Checklist—Attention/Aggressive/Anxious; CBCL-SED—severe emotional dysregulated profile; CBCL-DESR—deficient emotional dysregulation profile; CBCL-non-ED—not emotional dysregulated profile.

**Table 4 jcm-11-02922-t004:** Change in CBCL-AAA subscale scores and total scores over 3 months MPH treatment in the two groups.

CBCL Scores	ADHD Group Mean (SD)	ADHD and ASD Group Mean (SD)	*p* Value	Total Sample
**Subscales**				
**T0**				
Anxious	65.4 (7.6)	63.2 (10.9)	0.32	64.5 (9.1)
Attention	69.8 (7.8)	68.1 (10.3)	0.42	69.09 (8.9)
Aggressive	67 (11)	63.8 (13.1)	0.27	65.69 (11.9)
**T1**				
Anxious	60.8 (87.9)	58.8 (8.4)	0.31	60.01 (8.18)
Attention	61.3 (7.4)	57.9 (5.6)	0.04	59.89 (6.85)
Aggressive	58.9 (9.7)	57.3 (10.7)	0.29	58.79 (10.1)
***p*-value**	<0.01	0.01		<0.01
	<0.01	<0.01		<0.01
	<0.01	0.01		<0.01
Total score				
**T0**	202.2 (20.7)	195 (30.5)	0.24	199.27 (25.3)
**T1**	181.9 (20.2)	174 (21.4)	0.12	178.69 (20.9)
**DIFF T0–T1**	20.3	21	0.87	<0.01

CBCL—Childhood Behavior Checklist; ADHD—attention-deficit/hyperactivity disorder; ASD—autism spectrum disorder; SD—standard deviation.

**Table 5 jcm-11-02922-t005:** Side effects.

	ADHD Group *N* (%)	ADHD and ASD Group *N* (%)	*p*-Value
Loss of appetite	19 (46.34)	13 (44.83)	0.90
Abdominal discomfort	10 (24.39)	10 (34.48)	0.49
Headache	7 (17.07)	5 (17.24)	1.00
Palpitation	4 (9.76)	3 (10.34)	1.00
Irritability	1 (2.44)	4 (13.79)	0.15
Anxiety	3 (7.42)	2 (6.89)	1.00
Insomnia	2 (4.88)	0	0.51
Hyperfocusing	0	2 (6.89)	0.16

ADHD—attention-deficit/hyperactivity disorder; ASD—autism spectrum disorder.

## Data Availability

Not applicable.

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
