# Peer review of "Methylphenidate Use for Emotional Dysregulation in Children and Adolescents with ADHD and ADHD and ASD: A Naturalistic Study"

_jcm, 2022, doi:10.3390/jcm11102922_

Round 1
Reviewer 1 Report
I have no additional comments
Author Response
Thank you for your suggestions, we have revised our language.
Reviewer 2 Report
Please refer to the attached document for y comments.

Author Response
General comment: thank you for your clever consideration. We replaced efficacy with effectiveness throughout the manuscript whenever related to our naturalistic study.
Title: thank you for your improvement, we changed our title as suggested.
Abstract: thank you, we followed your minor corrections.
Results:
- We reported p-values in two decimal points as suggested.
- Thank you for your correction, we have corrected the sentence.
Discussion: we elaborated the recommended part as suggested (pag.13):
“Only two ASD patients with comorbid ADHD, both with intellectual disability, presented worsening of behavior with restlessness and increased stereotypes that caused the interruption of treatment and completely resolved after treatment discontinuation.
Previous review studies comparing MPH to placebo in children with ASD [63, 69] globally show that it was mildly superior for inattention, irritability and stereotypes, and did not worsen the core symptoms of ASD, and no adverse events were reported.
Even though the low number of individuals with intellectual disabilities in our sample did not allow us to draw conclusions about the role of intellectual disability on the effects of MPH, our findings are in accordance with previous studies [65] in which high functioning ASD were more likely to have a favorable response to MPH treatment in terms of efficacy and tolerability than low functioning ones and suggest greater caution in monitoring the effects of MPH in ASD with intellectual disability.”
As for study limitations we explained our choice of administering CBCL in the summer period in that paragraph (pag.14):
“The suitability of our study design used to measure ED administering CBCL during the summer medication-free period and after three months of treatment restart (employing each patient as his/her own control) could be discussed; future research in this field should involve proper administration of CBCL before the beginning of treatment.”
Conclusion:
- Yes, as we previously added in the discussion, “even though the low number of individuals with intellectual disabilities in our sample did not allow us to draw conclusions about the role of intellectual disability on the effects of MPH, our findings are in accordance with previous studies [65] in which high functioning ASD were more likely to have a favorable response to MPH treatment in terms of efficacy and tolerability than low functioning ones and suggest greater caution in monitoring the effects of MPH in ASD with intellectual disability.”
- Thank you for your suggestion, we changed the conclusion as recommended.

This manuscript is a resubmission of an earlier submission. The following is a list of the peer review reports and author responses from that submission.
Round 1
Reviewer 1 Report
Methylphenidate (MPH) use is very common in children with ASD for many years. This naturalistic study aims to compare the effect of MPH on Emotional Dysregulation (ED), a prominent symptom of both ADHD and ASD, in relatively small cohorts of children and adolescents with ADHD (n=41) and with high functioning ASD + ADHD (n=29).
I suggest the following corrections before reconsidering this MS for publication:
Title: OK
Abstract:
- "Nonetheless, research about it in ADHD children with comorbid Autism Spectrum Disorder (ASD) is still scant. Preliminary studies suggest that Methylphenidate (MPH) may be effective for ED in ADHD, while there is not enough evidence about its use in ADHD with comorbid ASD". This is not accurate. First the term ADHD children with comorbid ASD is not accurate. While ADHD might be a comorbidity of ASD, ASD is not a comorbidity of ADHD please modify to children with ASD and ADHD (here and all over the MS). More important, ED is a main symptom of both ADHD and ASD. After decades of extensive MPH use for both ADHD and ASD it is not accurate to say that only preliminary studies suggest its efficacy for ED.
- "This naturalistic study aims to investigate efficacy, using CBCL-AAA (Attention-Aggressive-Anxious), and safety of immediate and extended-release MPH in the treatment" . The study didn't assess safety (many previous studies did). If you meant negative effect on ED please modify to "This naturalistic study aims to investigate effect of immediate and.... on ED using CBCL-AAA (Attention-Aggressive-Anxious)"
- "Comparisons between groups were made by t-test or Wilcoxon test; comparisons of proportions have been performed by chi-square/Fisher exact test and by McNemar test. General linear model was applied to evaluate the difference at the two timepoints, with and without therapy". Detailed statistics do not belong here. Instead the authors should describe the study design. For example: parents of children.. completed the CBCL twice, once in the summer brake after a month without MPH treatment and again after 3 months of MPH treatment.
- "Results demonstrate that MPH is associated with a statistically significant reduction of ED in ADHD patients, also with comorbid ASD, without significant adverse events, supporting the use of psychostimulants for the treatment of ED in these neurodevelopmental disorders." please modify "also with comorbid ASD" to "and ADHD + ASD patients" and also modify "without significant adverse events" to "without substantial adverse events" (AS expected, there were adverse events).
Introduction: The authors should present former studies on MPH in children with ASD for example: Reichow et al., 2013 - a meta-analysis of four studies or: "Randomized, controlled, crossover trial of methylphenidate in pervasive developmental disorders with hyperactivity." Research Units on Pediatric Psychopharmacology Autism Network. Arch Gen Psychiatry. 2005 Nov; 62(11):1266-74
Methods:
The authors should describe the cohorts better: how was the diagnosis made "according to clinical judgment" is not enough. Did all the children fulfilled DSM-5 criteria and CPRS-R? can you present the scores? How ADHD severity was rated? (the authors declare that inclusion criteria was mod-severe ADHD. Based on what? CGI-S? ). How many of the ASD children had ADOS, ADI-R, and ASDI? can you present scores?
The design and ethic approvals are not clear. Was this a prospective or retrospective study? the authors declare that this study "was based on a clinical database" and "that all procedures, as well as follow-up and data collection were part of our standard routine" Are the IQ assessments and two CBCLs - 4 months apart included? Is it a routine to do a CBCL exactly in the middle of summer break? when did the parents signed consent? prior to initiation of treatment? prior to data collection? prior to first CBCL?
The authors included only participants who were treated for at least 3 months , excluding all those who had AEs and stopped the treatment earlier. Do you have the data on how many stopped the treatment earlier in each group? (this is highly important as previous studies suggest that children with ASD response less well to MPH and stop the treatment earlier and if we want to generalized the results to children with HF ASD and ADHD we have to be sure that the group presented here is representative ).
Please modify "summer break medication" page 6 line 118 . There is no such a term. This highly important data is not explained well. Do you mean that all the children in your cohorts didn't receive MPH during the entire summer break even though they have ED and MPH helped them for that? Is it common in your clinic? if so please state that here and discuss it in the discussion
Results:
The results are sparse (just two CBCLs) but presented in many tables and figures. Table 4 and 5 can be easily combined.
Figures 1-3 add no novel information and do not explain anything (it easier to understand without them)
Safety - There is plenty of data about safety of MPH and this study was not powered to assess safety. it will be more accurate to change it to tolerability - to compare tolerability between cohorts
Also, what do you mean by level 3 ASD , level 2 ASD? (according tp DSM? Requiring (very) substantial support?). And when did they stopped the MPH? after more than 3 months? (otherwise they shouldn't be included). Please specify. Where they the only ones who presented worsening of behavior? This is unlikely and if many more stopped the treatment earlier it should be mentioned as well
Discussion:
Please make clear that the ASD population in this study is high functioning (normal IQ and very few on anti-psychotics)
Study limitations are missing. Please include a paragraph of study limitations including mainly: small samples sizes, lack of randomization or blinding, lack of controlling for other pharmacological and behavioral treatments and the bias of assessing CBCL at two different scenarios summer break and school routine. It would be more accurate to compare the CBCLs before and after treatment initiation
Reviewer 2 Report
Overall, this manuscript reflects a good attempt of a simple pharmacotherapeutic research for children with ADHD and ADHD/ASD. However, there are many areas that the authors must improve.
I provide here the general comments. Please refer to the attached comments within the manuscript in pdf form.
- The manuscript appears unstructured. The sociodemographic data should be placed within the results. Furthermore, adding subsections for Sampling method and/or ethical considerations would help to presents the Methods better.
- The overall English is comprehensible but I suggest proof reading.
- Please elaborate on the test done to evaluate the equivalence, in the Styatistical Analysis. Also, the software used should be mentioned.
- Results - Please explain the need to analyse the total sample, rather than focusing on comparing the groups? Justify the need to analyse the total scores, since the subscales have been examined.
- Formatting of the table must conformed APA or the journal's.
